# Systemic metabolic depletion of gut microbiome undermines responsiveness to melanoma immunotherapy

Natalia V Zakharevich[1], Maxim D Morozov[1], Vera A Kanaeva[1,3], Mikhail S Filippov[4], Tatyana I Zyubko[4], Artem B Ivanov[1,2], Vladimir I Ulyantsev[2], Ksenia M Klimina[1], Evgenii I Olekhnovich[1]

Immunotherapy has proven to be a boon for patients battling metastatic melanoma, significantly improving their clinical condition and overall quality of life. A compelling link between the composition of the gut microbiome and the efficacy of immunotherapy has been established in both animal models and human patients. However, the precise biological mechanisms by which gut microbes influence treatment outcomes remain poorly understood. Using a robust dataset of 680 fecal metagenomes from melanoma patients, a detailed catalog of metagenome-assembled genomes (MAGs) was constructed to explore the compositional and functional properties of the gut microbiome. Our study uncovered significant findings that deepen the understanding of the intricate relationship between gut microbes and the efficacy of melanoma immunotherapy. In particular, we discovered the specific metagenomic profile of patients with favorable treatment outcomes, characterized by a prevalence of MAGs with increased overall metabolic potential and proficiency in polysaccharide utilization, along with those responsible for cobalamin and amino acid production. Furthermore, our investigation of the biosynthetic pathways of short-chain fatty acids, known for their immunomodulatory role, revealed a differential abundance of these pathways among the specific MAGs. Among others, the cobalamin-dependent Wood–Ljungdahl pathway of acetate synthesis was directly associated with responsiveness to melanoma immunotherapy.

## Introduction

Cutaneous melanoma is a type of skin cancer that has become increasingly common in recent decades. It is the 17th most common cancer worldwide, the 13th most common cancer in men, and the 15th most common cancer in women, according to the World Cancer Research Fund (https://www.wcrf.org). Despite the increasing incidence of the disease, survival and quality of life for patients have been significantly improved by novel approaches and tailored drugs (Switzer et al, 2022). Immune checkpoint inhibitors (ICTs) have made significant advances, resulting in durable remissions in more than 50% of patients with metastatic melanoma (Larkin et al, 2015). However, treatment can be associated with side effects such as dermatitis, colitis, hepatitis, antibody-related thyroid dysfunction, and in some cases, pneumonia (Horvat et al, 2015; Roy & Trinchieri, 2017; Wolchok et al, 2017; Robert et al, 2019). Studies are underway to identify specific host or tumor characteristics that may serve as predictors of patient response to ICT therapy, thereby improving immunotherapy outcomes.

The influence of the human gut microbiota on the efficacy of immunotherapy against malignant tumors is being studied extensively by the international scientific community. Studies in animal models have provided initial evidence that gut microbes may play a role in shaping the effects of anticancer therapies and in the development of antitumor immunity (Iida et al, 2013; Viaud et al, 2013; Sivan et al, 2015; Vétizou et al, 2015). These findings have subsequently been validated in studies of melanoma patients undergoing immunotherapy (Chaput et al, 2017; Frankel et al, 2017; Routy et al, 2018). Notably, the intestine microbiome has been found to be associated not only with the ICT response but also with the incidence of related side effects (Dubin et al, 2016; Li et al, 2019). These important findings described above have been further substantiated by fecal microbiota transplantation (FMT) into gnotobiotic mice (Gopalakrishnan et al, 2018; Matson et al, 2018; Lam et al, 2021) and clinical patients (Baruch et al, 2021; Davar et al, 2021; Routy et al, 2023).

The results outlined above have not only demonstrated how specific characteristics of the human gut microbiota influence immunotherapy outcomes but have also opened up the intriguing possibility of their transferability. The phenomenon of transmission of the responder phenotype by "fecal matter" suggests the

[1]Lopukhin Federal Research and Clinical Center of Physical-Chemical Medicine of Federal Medical Biological Agency, Moscow, Russian    [2]ITMO University, Saint Petersburg, Russian    [3]Moscow Institute of Physics and Technology, Moscow, Russian    [4]Bioinformatics Institute, Saint Petersburg, Russian

Correspondence: jeniaole01@gmail.com

potential involvement of specific microbial species, a combination of species, or microbial derivatives that can be isolated and used as adjuvants to enhance the efficacy of immunotherapeutic treatments. However, despite the large number of published studies and meta-analyses on this topic, the global scientific community still struggles with an incomplete understanding of the complex biological mechanisms underlying gut microbial regulation of the immune system in the context of cancer immunotherapy (Limeta et al, 2020; Lee et al, 2022).

Recently, Olekhnovich et al (2023) identified consistent stool metagenomic biomarkers associated with melanoma immunotherapy efficacy using community-accepted methods of taxonomic and functional annotation. In contrast, this study used advanced bioinformatics techniques such as genome-resolved metagenomics, strain profiling, comparative genomics, and metabolic reconstruction to refine and develop the proposed concepts. Together, these efforts aim to better understand the biological mechanisms underlying the influence of the gut microbiota on the regulation of antitumor immunity.

# Results

## Assembly of a nonredundant catalog of metagenome-assembled genomes (MAGs) using the melanoma patient metagenomes

In this study, we analyze patient fecal metagenomes collected before ICT administration from five previous studies (Frankel et al, 2017; Gopalakrishnan et al, 2018; Matson et al, 2018; Spencer et al, 2021; Lee et al, 2022). In addition, we present data demonstrating the positive impact of FMT on immunotherapy treatment outcomes (Baruch et al, 2021; Davar et al, 2021). In summary, this analysis includes stool metagenomes from a total of 680 individuals, consisting of 374 responders (R) and 306 nonresponders (NR) from seven studies. A comprehensive overview of the general characteristics of the dataset can be found in our previously published article (Olekhnovich et al, 2023). Fig 1 shows a schematic of the data analysis workflow.

MAGs were constructed for each sample using the provided data. This resulted in a total of 12,449 MAGs, which were then dereplicated to 98% average nucleotide identity. The final set consisted of 1,422 nonredundant MAGs with quality metrics of 93.3 ± 6.4 completeness and 1.6 ± 2.0 contamination. Fig S1 shows additional metrics such as N50 and assembly length. The resulting catalog conforms to the quality standards established by the Genomic Standards Consortium criteria (Bowers et al, 2017), with 1,006 high-quality (~71%) and 416 medium-quality (~29%) MAGs. According to the Genome Taxonomy Database classifications, the list of MAGs contains a total of 1,416 bacterial and six archaeal genomes, grouped into 13 different phyla. The most abundant phyla are Firmicutes (902 MAGs, ~63%), Actinobacteria (261 MAGs, ~18%), Bacteroidetes (148 MAGs, ~10%), Proteobacteria (59 MAGs, ~4%), and others (52 MAGs, ~4%). Table S1 provides overall assembly statistics and taxonomic annotation for the MAGs catalog. Fig 2A shows the phylogenetic tree constructed using 1,422 MAGs sequences.

The following step in our analysis involved obtaining relative abundance profiles of MAGs through the inStrain approach (Olm et al, 2021). The mapping results are presented in Table S2 and Fig 2B. The obtained profiles were used to assess changes of routine microbial ecology metrics, including alpha and beta diversity. In summary, alpha diversity statistically significantly depended on the dataset variable but not on the immunotherapy outcome (ANOVA, adj. $P < 0.001$), whereas beta diversity depended on both examined variables (PERMANOVA, adj. $P < 0.001$; Table S3 and Fig 2C–E). To identify the specific bacteria specifically associated with responsiveness to immunotherapy, a data analysis strategy outlined in the following section was pursued.

## Identify differences between R and NR groups across datasets

Metagenomic data are compositional, which limits the use of statistical methods directly without any transformations (Gloor et al, 2017). A number of methods have been developed to represent compositional data in the Cartesian space. In this study, the following data analysis protocol was used. The Songbird approach was used to generate the ranking differentials, which describes the log fold change of MAGs' relative abundance associated with the immunotherapy outcome variable (Morton et al, 2019). It is important to note that Songbird does not provide $P$-values, making it difficult to estimate statistical significance using this approach alone. To overcome this limitation, the Qurro method (Fedarko et al, 2020) was used to calculate log ratios based on the ranked MAG features. Standard statistical methods can estimate obtained log ratio values that condense multiple microbial traits into a single value, similar to alpha diversity indexes. This method is useful for ecological modeling and statistical evaluation because it allows results to be interpreted in the context of ecological "states" without requiring separate hypothesis tests for each MAG. It also allows tracking changes in microbiome composition over time, facilitating the identification of transitions between different ecological "states."

Identification of MAGs associated with immunotherapy outcomes was performed according to the outlined protocol. Using the Songbird approach, genomes associated with R and NR groups were identified individually for each dataset. MAGs with an absolute differential value > 0.3 were selected for further analysis. As a result, the log ratios of the relative abundances of the selected MAGs showed a clear statistically significant difference (see Fig 3A; Wilcoxon rank-sum test adj. $P < 0.001$). Furthermore, the calculated log ratios depended on both response and donor variables in the FMT datasets (Table S3; ANOVA, adj. $P < 0.01$). FMT responders showed statistically significant increased log ratio values compared with FMT nonresponders, which was confirmed by additional statistical tests (Wilcoxon rank-sum test Baruch et al, 2021 adj. $P < 0.001$, Davar et al, 2021 adj. $P < 0.001$). The log ratio–based measure presented shows the state of the microbiome in the context of immunotherapy and assesses the evolution of recipient samples over time (Fig 3B). In addition, FMT responders had statistically significantly increased log ratio values before fecal transfer compared with FMT nonresponders. Notably, this effect was reproducible in both FMT datasets (Wilcoxon rank-sum test, for Baruch et al, 2021 adj. $P < 0.01$, for Davar et al, 2021 adj. $P < 0.01$).

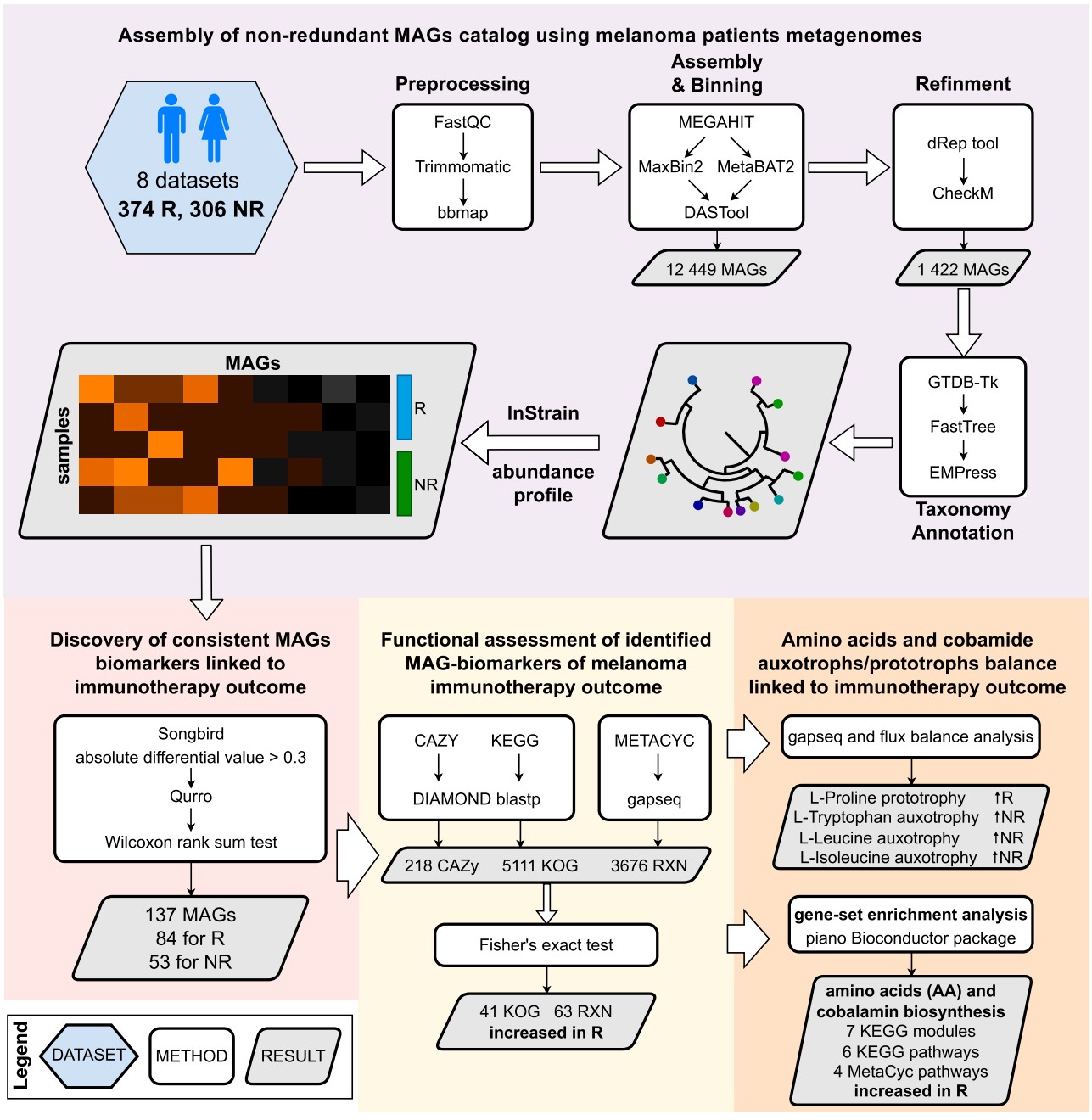

**Figure 1. Data analysis workflow.**

## Discovery of consistent MAG biomarkers linked to the immunotherapy outcome

Using the genome sets identified in the previous analysis step, a list of 137 consistent MAG biomarkers was identified (see the Materials and Methods section). Of these, 84 MAGs were associated with positive immunotherapy outcomes (R biomarkers), whereas 53 were associated with negative outcomes (NR biomarkers). These MAGs belonged to six phyla with the following distribution: Firmicutes (38 negative, 65 positive), Bacteroidetes (7 negative, 9 positive), Actinobacteria (1 negative, 8 positive), Proteobacteria (5 negative, 1 positive),

Verrucomicrobiota (2 negative, 0 positive), Desulfobacterota (0 negative, 1 positive) (Table S4). Notably, five MAGs—including two *Bifidobacterium adolescentis*, one unclassified *Bifidobacterium*, *Gemmiger qucibialis*, and *Barnesiella intestinihominis*—identified as R biomarkers in six studies. In contrast, NR biomarkers, such as *Akkermansia* sp004167605 and *Scatavimonas* sp900540275, were reproducible in no more than four datasets. The phylogenetic tree generated by OrthoFinder using all MAGs sequences is shown in Fig 3C. The obtained biomarker sets were further validated using machine learning methods.

The application of machine learning models using stool metagenomes to predict and/or classify various human disease states

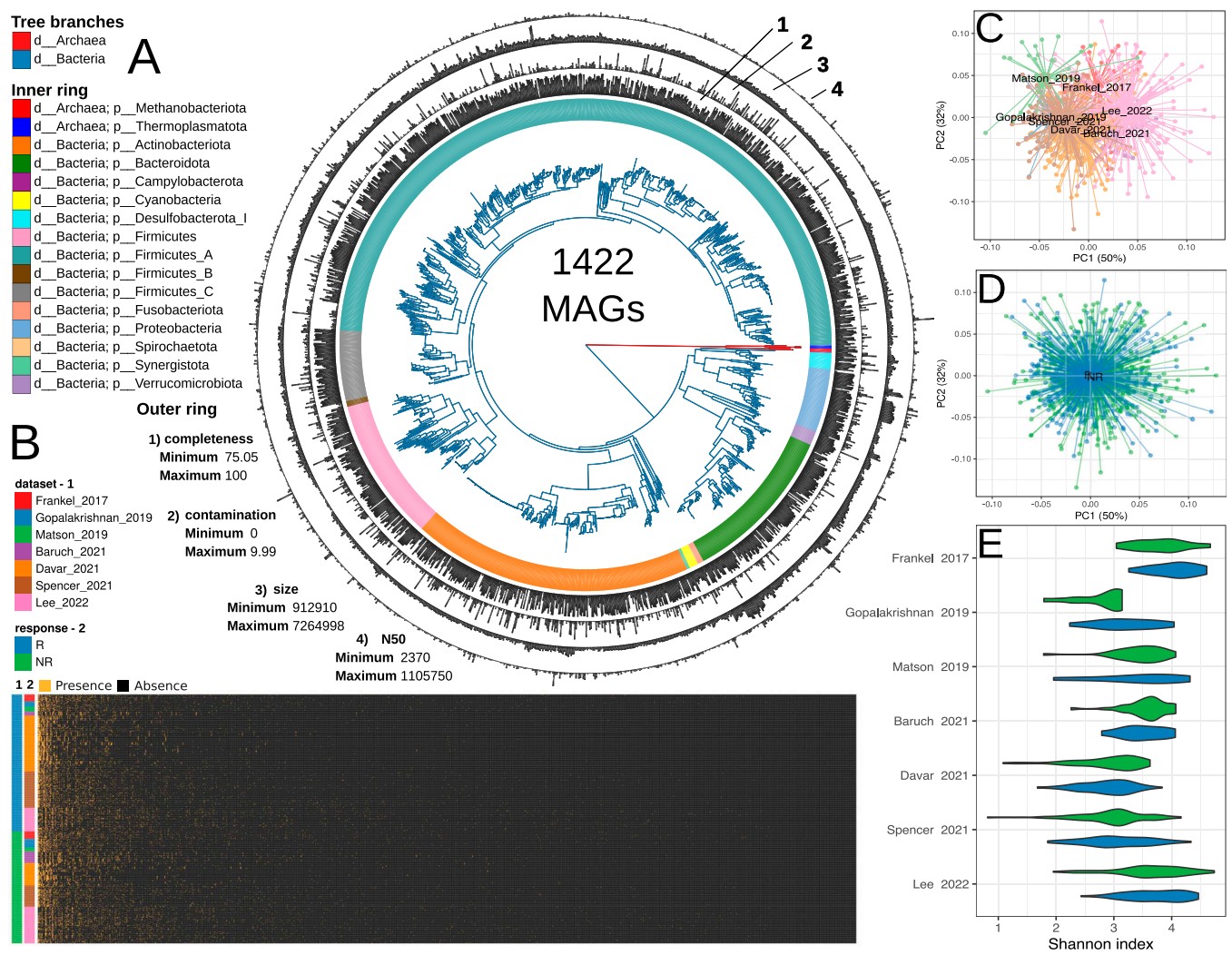

**Figure 2. Summary of metagenome-assembled genome (MAG) catalog assembly, taxonomic annotation, metagenomic sample mapping, and basic metagenomic analysis.**
**(A)** Approximate maximum likelihood phylogenetic tree generated using CheckM with 43 AA marker sequences and 1,422 MAGs assembled from 680 melanoma patient stool metagenomes. Branches are color-coded according to bacterial or archaeal affiliation. The inner ring shows phylum-level taxonomic annotations aligned with the phylogenetic tree, whereas the outer ring shows MAGs assembly statistics. **(B)** Heatmap illustrating the results of mapping metagenomic reads to the MAG catalog using the inStrain tool. The color bars on the left indicate the datasets (1) and the response variables (2). MAGs' presence/absence are plotted on the x-axis, whereas stool samples are plotted on the y-axis. The presence of MAG in the samples is indicated by color: yellow—the MAG is present in the sample, black—the MAG is absent in the sample. **(C, D)** Multidimensional scaling biplot showing the relative abundance profiles of MAGs in stool metagenomes from different studies (C) and with different immunotherapy outcomes (D). Samples are represented by dots connected to the centroid. Data set and immunotherapy response variables are shown in different colors. The color scheme corresponds to the legend of figure (B). **(E)** Shannon index values representing the diversity of MAG relative abundance profiles of stool metagenomes, stratified by dataset and the immunotherapy outcome. The color scheme corresponds to the legend of Figure (B).

has not yet been widely adopted in clinical practice. This may be because of insufficient sample size for model training and the overall complexity of the data, which is characterized by sparsity and high inter-individual variability. In addition, combining data-sets to improve classification quality is complicated by the "batch effect," that is, the classification quality on an independent dataset not used in training is likely to be unsatisfactory. In this case of the melanoma immunotherapy data, a random forest (RF) classifier using MAG relative abundance values directly did not yield re-producible predictions between datasets according to out-of-dataset cross-validation (six datasets used for training, one used for testing), supporting the above thesis (ROC AUC = 0.54 ± 0.17; Fig

S2A). However, using log ratios obtained using MAGs with absolute differential values > 0.3 significantly improves prediction (633 out of 680 samples; ROC AUC = 0.91 ± 0.06, Wilcoxon rank-sum test P = 0.001; Fig S2B). Obviously, the interpretation of this model is challenging because its use of specific MAG features to classify each dataset. We can assume that the objective biological differ-ence between the R and NR groups within each dataset is described by a different set of features, united by a similar biological meaning. However, the practical usefulness of such a model is questionable. Perhaps the log ratios determined on the basis of representative sets of features common to all datasets will help to solve this problem. It should be noted that NR biomarkers cannot be a good

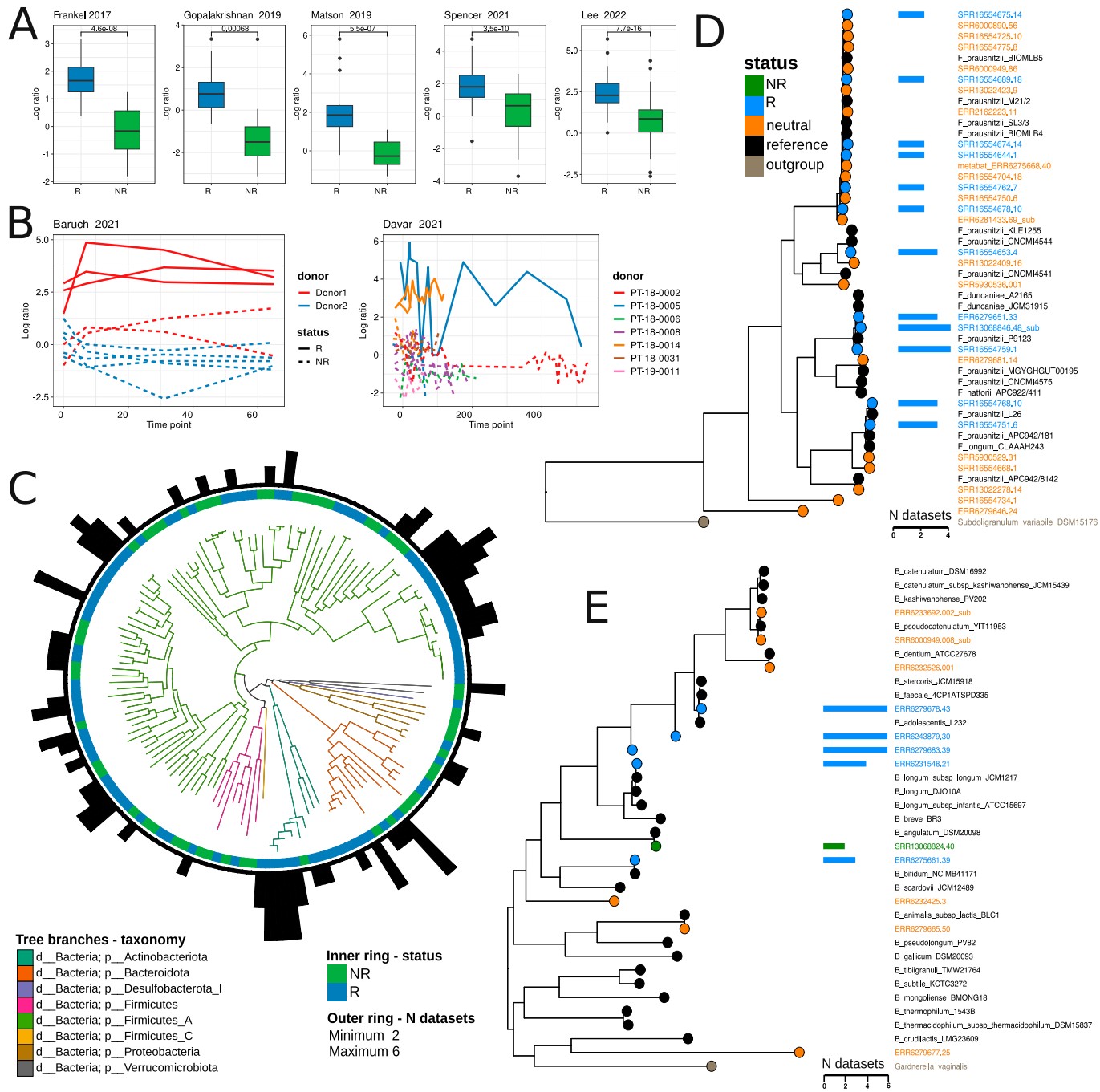

**Figure 3. Metagenome-assembled genome (MAG) biomarker discovery and characterization.**
**(A)** Log ratio plots of selected feature data obtained with Songbird and Qurro software using non-fecal microbiota transplantation datasets. MAG features with a differential value > 0.3 were selected as the numerator, whereas MAGs with a differential value < −0.3 were selected as the denominator for the log ratio calculation. Statistical evaluation of log ratios using the Wilcoxon rank-sum test shows significant differences between the R and NR groups, with unadjusted *P*-values shown. All adjusted *P*-values were < 0.001. **(B)** Log ratio plots of selected feature data obtained with Songbird and Qurro software using fecal microbiota transplantation datasets. Log ratio values were clustered on the recipient variable and plotted according to time points: one line corresponds to one recipient. The recipient's lines are colored according to their affiliation with a particular donor. **(C)** Phylogenetic tree based on AA sequences of identified MAG biomarkers obtained with OrthoFinder. Tree branches are color-coded according to taxonomic annotations at the phylum level. The inner ring links MAG biomarkers to the R or NR group, whereas the outer ring indicates the dataset numbers where the biomarker was discovered. **(D, E)** OrthoFinder-generated phylogenetic trees based on *Faecalibacterium* (D) or *Bifidobacterium* (E) MAGs and references. Leaf colors correspond to different genome sets, and additional bar graphs show the dataset numbers where the biomarker was found.

choice for constructing log ratios because they cannot characterize a meaningful number of samples (491 out of 680). Using bacterial features with <−0.3 Songbird differential specific to each dataset

and 84 common R biomarkers allows the quality of prediction to remain as high (630 out of 680 samples; ROC AUC = 0.89 ± 0.09, Wilcoxon rank-sum test *P* = 0.71; Fig S2C). The results obtained may

indicate the presence of bacteria that are more common in R patients but not in NR patients. We have previously shown that the common feature of NR patients may be the presence of opportunistic species in the stool metagenome (Olekhnovich et al, 2023). Obviously, the set of these abnormal species may be different in each case.

### Strain-specific features of *Faecalibacterium* and *Bifidobacterium* MAGs linked to the melanoma immunotherapy outcome

Interestingly, 30 out of 1,422 MAGs with the level of nucleotide identity <98% were taxonomically annotated as *Faecalibacterium* spp. Eleven *Faecalibacterium prausnitzii* strains were associated with positive immunotherapy outcomes in six out of seven datasets, whereas 19 had a neutral status according to the biomarker discovery protocol. To gain deeper insights, the phylogenetic tree encompassing all 30 *Faecalibacterium* MAGs was reconstructed along with reference genomes of *Faecalibacterium* species inhabiting the human gut. As an outgroup, we included the genome sequence of *Subdoligranulum variabile* DSM 15176. The results are shown in Fig 3D. All MAGs, including the 11 biomarker MAGs, were distributed among different clades within the tree. This could be attributed to the high plasticity of the genomes of *F. prausnitzii* species, suggesting that these MAGs likely belong to different phylogroups. Furthermore, our phylogenetic analysis revealed that three R biomarker MAGs, namely, SRR13068846.48_sub, SRR16554759.1, and ERR6279651.33 belong to the species *Faecalibacterium duncaniae* (strain *F. prausnitzii* P9123, which despite its name belongs to the *F. duncaniae* group (Tanno et al, 2022). It is noteworthy that only the *F. duncaniae* clade did not contain neutral MAG biomarkers.

The final set of nonredundant MAGs included 12 MAGs assigned to the *Bifidobacterium* genus. These MAGs showed different associations with immunotherapy outcomes. Specifically, five of them were classified as R biomarkers (ERR6279678.43, ERR6243879.30, ERR6279683.39, ERR6231548.21, and ERR6275661.39), one (SRR13068824.40) as a NR biomarker, whereas the remaining six were not included in the list of 137 MAG biomarkers. The taxonomic classification of these six MAGs is as follows: ERR6279678.43 and ERR6243879.30 were classified as *B. adolescentis*, ERR6231548.21 as *Bifidobacterium longum*, ERR6275661.39 as *Bifidobacterium bifidum*, SRR13068824.40 as *Bifidobacterium angulatum*, and ERR6279683.39 was assigned to the *Bifidobacterium* spp. without clear species annotation. To clarify the species identity of ERR6279683.39 MAG, the phylogenetic tree was reconstructed including all 12 MAGs and reference genomes of *Bifidobacterium* species inhabiting the human gut. As an outgroup, we used the genome sequence of *Gardnerella vaginalis* UMB0386 and obtained the results shown in Fig 3E. The analysis revealed that MAG ERR6279683.39 and, unexpectedly, MAG ERR6243879.30 occupied positions on the tree between branches related to the *B. adolescentis* group and the *B. longum* group. This observation prompted us to further test the bifidobacterial MAGs for chimeric assembly by GUNC (Orakov et al, 2021). Although both MAGs passed the test based on "pass.GUNC" in the output file, a closer examination of the output files in the "gene_calls" and "diamond_output" folders revealed that for MAG ERR6279683.39, 695 *genes* were assigned to the *B. longum*, and almost the same number, 571

*genes*, were assigned to the *B. adolescentis*. Based on this, we believe that MAG ERR6279683.39 may indeed be a chimeric MAG, which probably explains its intermediate position on the tree between two species. As for MAG ERR6243879.30, there were 895 *genes* assigned to the *B. adolescentis* and 237 *genes* assigned to the *B. longum*. This indicates a possible contamination in this MAG, which could explain its placement outside the branch of the *B. adolescentis* group.

### Functional assessment of MAG biomarkers of the melanoma immunotherapy outcome

The annotation of the MAG biomarkers involved the use of various functional databases, including CAZy (carbohydrate-active enzymes, http://www.cazy.org), KEGG (Kyoto Encyclopedia of Genes and Genomes, https://www.genome.jp/kegg), and MetaCyc (https://metacyc.org). This comprehensive annotation effort resulted in the assignment of 218 CAZy categories, 5,111 KEGG orthologous groups (KOG), and 3,676 MetaCyc Reactions (RXN). Derived functional profiles are available in Tables S5, S6, and S7. PERMANOVA analysis was performed to understand the relationship between the gene categories in MAG biomarkers and the phylum and immunotherapy response variables. The results indicated that the abundance of all studied gene categories in MAG biomarkers was linked to both the phylum and immunotherapy response variables. Specifically, the content of KOG and RXN were significantly linked to the phylum and response variables, whereas CAZy categories profiles were also linked to the phylum, but the relationship with the response variable was at a lower significance level. Detailed results of these analyses are presented in Table S8.

Additional statistical tests showed that the abundance of the KEGG and MetaCyc gene groups increased in the R biomarkers. However, there were no significant changes in the CAZy categories in any of the biomarker groups (Fig S3). Notably, the Bacteroidetes MAGs tended to increase the number of CAZy categories in the R group (Fig S4; Wilcoxon rank-sum test, adj. $P$ = 0.07). In addition, the Bacteroidetes MAGs R group showed an enrichment in glycoside hydrolase (GH) families compared with NR (Wilcoxon rank-sum test, $P$ = 0.006). Specifically, only R biomarkers such as *Bacteroides ovatus* (N CAZy = 123; GH = 70), *Bacteroides xylanisolvens* (N CAZy = 109; GH = 64), and *Bacteroides uniformis* (N CAZy = 92) were observed. Among the top five MAGs with the highest number of CAZy categories and GH families were *B. ovatus* (N CAZy = 78; GH = 56), *Bacteroides nordii* (N CAZy = 74; GH = 41), and *Parabacteroides distasonis* (N CAZy = 73; GH = 45). The complete list of Bacteroidetes MAGs containing CAZy categories can be found in Table S9. Furthermore, when analyzing the number of genes at the phylum level, only Firmicutes and Bacteroidetes showed a statistically significant difference in the number of KEGG and MetaCyc gene groups, as shown in Fig S4. In addition, Fig S5 shows a two-dimensional visualization based on nonmetric multidimensional scaling of functional profiles.

Using Fisher's exact test and applying false discovery rate (FDR) corrections for multiple testing, we successfully identified specific gene groups that distinguish functional categories among MAG biomarkers. Specifically, we found 41 KOG and 63 RXN categories

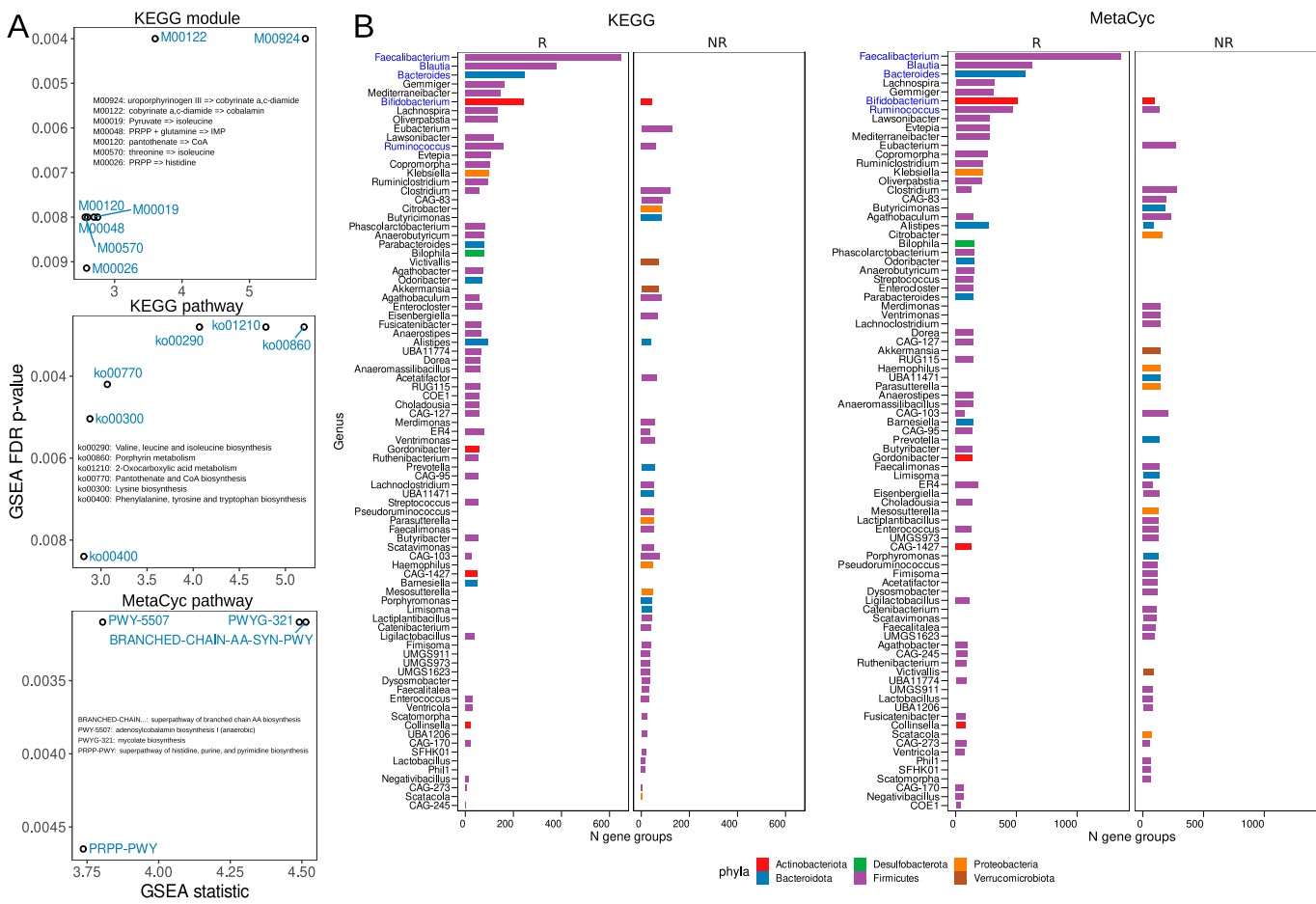

**Figure 4. Functional differences between metagenome-assembled genome biomarker groups.**
**(A)** Gene Set Enrichment Analysis (GSEA) results, where the x-axis represents GSEA analysis statistics and the y-axis represents false discovery rate–adjusted *P*-values for identified functional categories. GSEA statistic values take only positive values because identified pathways are only associated with immunotherapy response (R group) but not with negative outcome (NR group). The metabolic pathway names are transcribed within the figures. **(B)** Bacterial genera containing genes associated with differentially abundant KEGG and MetaCyc functional pathways. The x-axis indicates the total number of defined gene groups, whereas the y-axis corresponds to bacterial genera. Genera belonging to bacterial phyla are highlighted in color.

that showed significant differences (Table S10; adj. *P* threshold < 0.05). Notably, all of these identified gene groups were up-regulated in the R biomarkers. To gain an understanding of the pathways distinguishing different biomarker groups, we performed gene set enrichment analysis (GSEA). The results of this analysis revealed that seven KEGG modules, six KEGG pathways, and four MetaCyc pathways associated with amino acid (AA) and cobalamin biosynthesis were significantly up-regulated in the R biomarker group (see Fig 4A). It is worth noting that according to MetaCyc-based GSEA analysis, the PWYG-321 mycolate biosynthesis pathway appeared to be up-regulated in the R group. Interestingly, mycolate is an exclusive component of the cell wall of mycobacteria. We further investigated this finding and translated the PWYG-321–related reactions (RXNs) into the Enzyme Commission (EC) nomenclature, followed by mapping to the KEGG database. This analysis revealed that the resulting ECs were related to ko00061: fatty acid biosynthesis pathway (Fig S6). Thus, we considered the initial observation related to the mycolate biosynthesis pathway to be an artifact of the analysis.

In addition, we explored the relationship between MAG biomarkers and the aforementioned immunotherapy-relevant pathways. The results of this analysis, shown in Fig 4B, highlighted the top five genera that contained the highest number of gene clusters from these pathways. These genera were *Faecalibacterium*, *Blautia*, *Bacteroides*, *Bifidobacterium*, and *Ruminococcus*.

### Amino acid and cobalamin auxotroph/prototroph balance linked to the immunotherapy outcome

From our results, it is clear that the pathways related to the biosynthesis of AAs and cobalamin show consistent GSEA results across different gene sets. However, what piques our interest is to explore the contribution of both producers (prototrophs) and consumers (auxotrophs) of these vital compounds to the outcome of melanoma immunotherapy. Our research used gapseq and flux balance analysis to identify AA prototrophs and auxotrophs. The list of target AAs included L-arginine, L-asparagine, L-cysteine, L-glutamine, L-histidine, L-isoleucine, L-leucine, L-lysine, L-methionine,

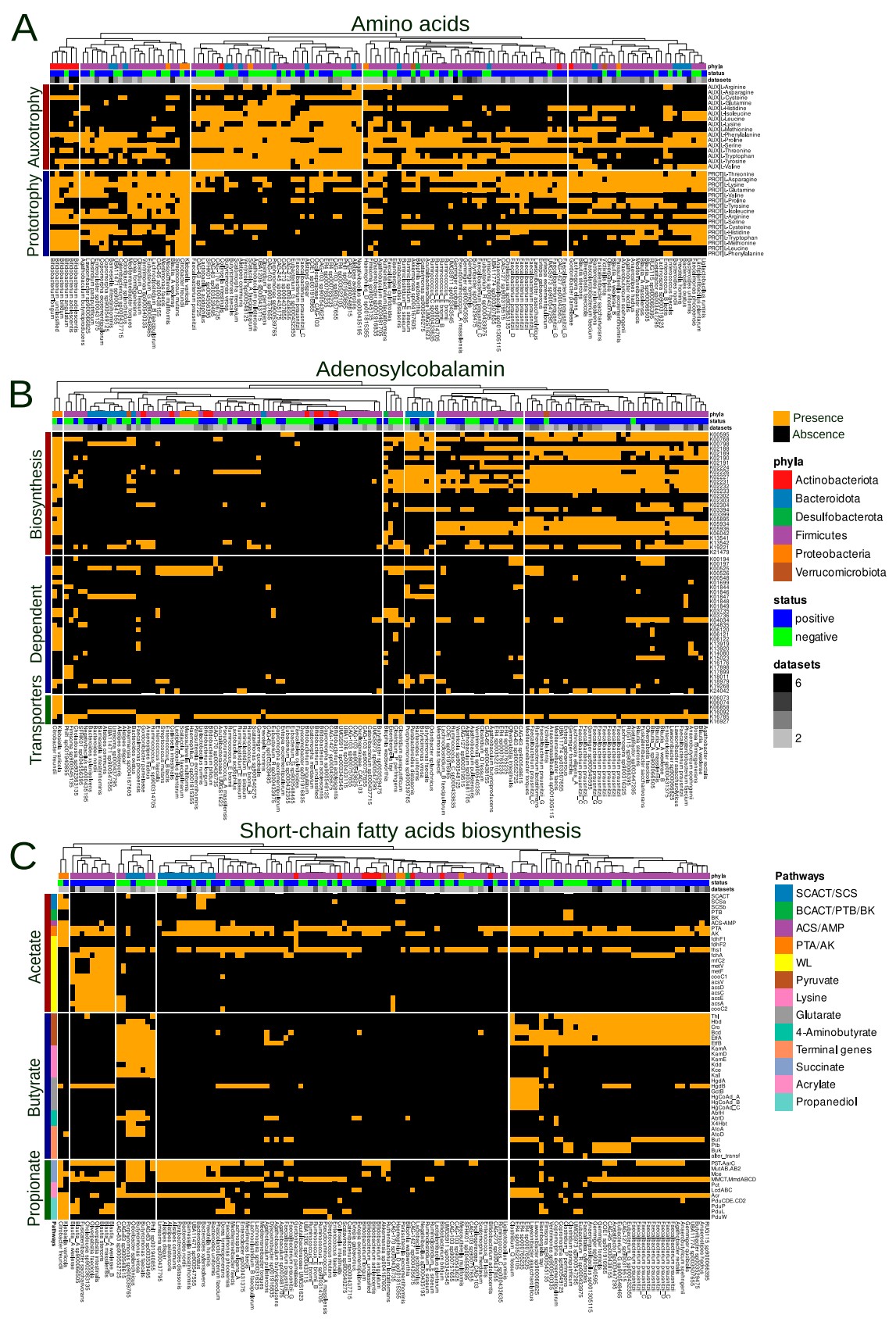

L-phenylalanine, L-proline, L-serine, L-threonine, L-tryptophan, L-tyrosine, and L-valine. Notably, our PERMANOVA results revealed a statistically significant association between the distribution of AA prototrophs and auxotrophs and both strain and response variables (adj $P$ < 0.01; Table S11). Further statistical analysis revealed that the frequency of AA prototrophy events was higher in positive immunotherapy outcomes, whereas AA auxotrophy events were more frequent in negative immunotherapy outcomes (Wilcoxon rank-sum test, adj. $P$ = 0.01; Fig S7A). Fisher exact tests were used to identify specific auxotrophy/prototrophy events associated with different MAG biomarker groups. The results obtained indicate that L-proline prototrophy is significantly increased only in R biomarkers, whereas L-tryptophan, L-leucine, and L-isoleucine auxotrophy are significantly increased in NR biomarkers. At lower significance levels, this trend persists, particularly with the absence of increased AA auxotrophy in R biomarkers and no increase in prototrophy in NR biomarkers (Table S11). A visual representation of the distribution of AA auxotrophic/prototrophic events is shown in Fig 5A.

As gapseq did not identify the cobalamin biosynthesis pathways within the biomarker MAG sets, KOGs belonging to the cobalamin biosynthesis KEGG modules (M00924, M00122) were selected for further analysis. In addition, gene groups encoding cobalamin-dependent enzymes (N = 40) and transporters (N = 6) were included in our analysis. According to the ANOVA, the distribution of cobalamin-related genes (biosynthesis, dependent, and transporters) was subsequently associated with both strain and immunotherapy response variables (Table S12). Further statistical analysis revealed an increase in the number of cobalamin biosynthesis genes in the R biomarkers, whereas the number of genes encoding cobalamin-dependent enzymes remained unchanged between the different MAG biomarker groups (Fig S7B). Fisher's exact test results indicated an increase in the occurrence of 15 cobalamin biosynthesis genes and 1 cobalamin-dependent enzyme gene in positive MAG biomarkers (Table S12). By mapping to the KEGG database, we identified KOGs associated with the cobalamin biosynthesis modules M00924 and M00122 (Fig S8). A visual representation of the distribution of cobalamin-related genes among the MAG biomarkers is shown in Fig 5B.

### Differential abundance of short-chain fatty acid (SCFA) biosynthesis pathways and their association with the immunotherapy outcome

A number of studies have shown that the SCFAs boost the overall immunity and improve the results of immunotherapy (Arpaia et al, 2013; Furusawa et al, 2013; Park et al, 2015; He et al, 2021; Luu et al, 2021). Therefore, it is reasonable to investigate variations in the content of SCFA biosynthesis pathways among MAG biomarkers. The initial assessment included the prediction of SCFA autotrophs/

phototrophs using gapseq and flux balance analysis methods. An ANOVA using PERMANOVA revealed that SCFA production was correlated with phyla but not with immunotherapy response variables (Table S13). Furthermore, Fisher's exact test revealed no statistically significant differences in the predicted producers of acetate, butyrate, and propionate between marker groups.

In particular, certain pathways responsible for acetate and propionate production require cobalamin. Therefore, it is valuable to examine the distribution of cobalamin-producing genes among SCFA producers from the MAG biomarker list. Our results indicate an increase in the number of cobalamin biosynthesis genes among the R biomarkers of acetate and butyrate producers compared with the NR biomarkers (Fig S9). However, no statistically significant difference in the number of cobalamin biosynthesis gene groups among the predicted propionate products was observed between the marker groups. The distribution of cobalamin biosynthesis genes among the SCFA producers is visually presented in Fig S10.

The results of the previous analyses were complemented by the catalog of predicted SCFA biosynthetic gene groups. We identified a total of 13 SCFA pathways, including five related to acetate biosynthesis, five to butyrate biosynthesis, and three to propionate biosynthesis. The presence of genes involved in SCFA biosynthetic pathways among the MAG biomarkers is shown in Fig 5C. According to PERMANOVA, the content of SCFA biosynthesis genes is statistically significantly associated with both phyla and immunotherapy response variables (Table 14). In addition, Fisher's exact test and further GSEA analysis revealed an association between the Wood–Ljungdahl pathway of acetate biosynthesis and MAG biomarkers of positive immunotherapy outcomes, whereas butyrate biosynthesis from the lysine pathway was associated with negative immunotherapy outcomes.

## Discussion

Understanding the biological mechanisms underlying the interactions between the immune system and the human gut microbiome is central to improving the efficacy of cancer immunotherapy. In contrast to other cancers, a sufficient amount of metagenomic data from melanoma patients receiving immunotherapy has been collected and is available in biological databases. By applying advanced bioinformatics techniques and reanalyzing data from these numerous research studies, we can gain deeper insights into the impact of gut microbial consortia on the regulation of antitumor immunity. In addition, such research efforts will broaden our understanding of the fundamental role of the microbiota in contributing to human health.

In our research, we used genome-resolved metagenomics with strain profiling, CoDa methods, and stool metagenomes from seven previously published studies to identify sets of microbes associated

**Figure 5. Distribution of AA auxotrophs/prototrophs and cobalamin/short-chain fatty acid biosynthesis genes in metagenome-assembled genome (MAG) biomarkers.**
**(A)** Bar graph showing the distribution of predicted auxotrophy/prototrophy for specific AAs or the number of gene groups involved in cobalamin biosynthesis. The bacterial genera are defined on the x-axis. **(B)** Distribution of cobalamin biosynthesis genes across MAG biomarkers, with the bacterial genera plotted on the x-axis. **(C)** Distribution of short-chain fatty acid biosynthesis genes among MAG biomarkers, stratified by specific pathways. The x-axis indicates bacterial genera, and the y-axis indicates genes involved in acetate, butyrate, or propionate biosynthesis.

with immunotherapy success. The list of 137 reproducible between datasets MAG biomarkers that distinguished patients based on the success of their immunotherapy were identified, according to the results of the analysis. Among the most consistently reproducible MAG biomarkers associated with positive immunotherapy outcomes were *B. adolescentis*, unclassified *Bifidobacterium*, *B. intestinihominis*, and *G. qucibialis*. It should be noted that several *Bifidobacterium* species, including *B. longum* and *B. bifidum*, have also been identified as markers of successful immunotherapy outcomes, albeit in fewer datasets. The existing scientific literature supports these findings, with published studies reporting *Bifidobacterium* species as indicators of favorable immunotherapy outcomes (Lee et al, 2021, 2022; Olekhnovich et al, 2023; Zhao et al, 2023). In addition, results from laboratory animals support these observations (Sivan et al, 2015; Lee et al, 2021; Yoon et al, 2021). In contrast, a meta-analysis conducted by Limeta et al (2020) was the only study to find an association between *B. intestinihominis* and improved immunotherapy outcomes. Interestingly, this bacterium has also been shown to enhance the effects of chemotherapy (Daillère et al, 2016) and vascular endothelial growth factor-tyrosine kinase inhibitor treatment (Dizman et al, 2021). In turn, *G. qucibialis* has previously been identified as beneficial for the positive outcome of immunotherapy in hepatobiliary cancer (Mao et al, 2021).

*F. prausnitzii* strains have also been identified as biomarkers of positive immunotherapy outcomes. According to numerous studies and meta-analyses, *F. prausnitzii* stimulates the immune system and improves the response to immunotherapy in several types of cancer (Chaput et al, 2017; Frankel et al, 2017; Gopalakrishnan et al, 2018; Matson et al, 2018; Peters et al, 2019; Limeta et al, 2020; Mao et al, 2021; Spencer et al, 2021; Olekhnovich et al, 2023). In the intestine, *F. prausnitzii* is one of the major producers of the SCFAs, including butyrate. Butyrate, a product of gut bacteria, enhances cytotoxic immunity and maximizes the results of immunotherapy, as shown in several studies (Bachem et al, 2019; Danne & Sokol, 2021; He et al, 2021; Luu et al, 2021). It is known that co-culturing *F. prausnitzii* with bifidobacteria increases colonization and promotes butyrate synthesis, probably because bifidobacteria can produce acetate, which *F. prausnitzii* needs for growth (Rios-Covian et al, 2015; Kim et al, 2020).

Investigation of the functional potential of MAG biomarker sets may provide insight into processes involving the microbiota that influence antitumor immunity. In addition, the implementation of genome-resolved metagenomics techniques allows the study of functions directly associated with specific genomes, improving the quality of analysis and facilitating interpretation. Initially, MAG biomarkers were evaluated using the CAZy database. High CAZy category counts in MAG Bacteroidetes have been associated with successful immunotherapy outcomes. Among the R biomarkers, *Bacteroides ovatus*, *Bacteroides xylanisolvens*, *Bacteroides uniformis*, *Bacteroides nordii*, and *P. distasonis* ranked in the top five MAGs with the highest amount of CAZy categories and GH families. Certain Bacteroidetes are known for their ability to break down glycans using thousands of different enzyme combinations (Lapébie et al, 2019). The utilization of polysaccharides by *Bacteroides*, namely, *B. uniformis*, has been shown to influence community dynamics and butyrate synthesis in another study (Feng

et al, 2022). In its turn, experiments on laboratory animals demonstrated the improving effect of immunotherapy results using dietary fiber (Spencer et al, 2021). This suggests that glycan digestion by *Bacteroides* may promote changes in the microbiota leading to increased butyrate synthesis (the immunomodulatory properties of which were outlined above), which may be associated with improved immunotherapy response.

Further in-depth functional analysis revealed that the group with R biomarkers had elevated levels of seven KEGG modules, six KEGG pathways, and four MetaCyc pathways related to the production of compounds necessary for immunity, including AA, medium- and long-chain fatty acids, and cobalamin. In addition, the list of the top five genera containing the most genes from these metabolic pathways included *Faecalibacterium*, *Blautia*, *Bacteroides*, *Bifidobacterium*, and *Ruminococcus*. The effect of AAs in supporting immune function has been extensively documented in various studies and does not require further detailed evidence or interpretation (Kelly & Pearce, 2020). Microbiome-produced medium- and long-chain fatty acids have the potential to stimulate antitumor immunity by binding to free fatty acid receptors. However, the links between cobalamin produced by gut microbes and human immunity are less straightforward. On the one hand, microbially derived cobalamin and other corrinoids may play an ecological role, being distributed by producers within the community and used by cobalamin-auxotrophic microbes (Degnan et al, 2014). On the other hand, corrinoids can be shared between microbes and Caco epithelial cells via vesicular transport (Juodeikis et al, 2022). Therefore, adding additional cobalamin from gut microbes as a supplement to the dietary form may potentially help the immune system's ability to fight tumors. In addition to functional potential analysis, the metabolic reconstruction approach revealed that AA and cobalamin prototrophs were associated with positive immunotherapy outcomes, whereas auxotrophs were associated with unfavorable outcomes. It is conceivable that "altruistic" bacterial behavior may make them more beneficial to the host and build community resilience.

Some studies have shown that SCFAs improve overall immunity (Arpaia et al, 2013; Furusawa et al, 2013; Park et al, 2015) and influence the outcome of immunotherapy (He et al, 2021; Luu et al, 2021). The MAG biomarker sets did not show significant differences in their predicted ability to produce acetate, butyrate, or propionate. However, the Wood–Ljungdahl (WL) acetate production pathway was associated with successful treatment outcomes, whereas the butyrate biosynthesis pathway via lysine degradation was associated with unsuccessful immunotherapy, according to the analysis of the reconstructed SCFA biosynthetic pathways. Notably, three bacterial genera, known to produce major SCFAs via the WL pathway (*Blautia*, *Fusicatenibacter*, and *Oliverpabstia*), are also involved in cobalamin biosynthesis (see Fig 5B). Notably, we did not find any "fdh" genes in *Blautia* genomes (see Fig 5C). Perhaps, in this case, *Blautia* is able to use the formate produced by the bifidobacteria (Rios-Covian et al, 2015; Trischler et al, 2022). The change in activity of the WL pathway may be driven by a cross-feeding relationship with *Bifidobacterium* species (e.g., *B. bifidum*) as they are specialized carbohydrate-fermenting species that produce the substrates for $CO_2$ fixation by the WL pathway (Plichta et al, 2016). In addition, the WL pathway shows a kind of positive feedback—it

provides additional acetate production, which in turn influences the increase in butyrate production (Koh et al, 2016). This strategy appears to be more advantageous because it does not use other important metabolites, in this case AA, for the synthesis of SCFA. In addition, acetate produced by the gut microbiota can directly improve immune function (Jugder et al, 2021; Yu et al, 2023).

Recently published studies showed the effectiveness of FMT received from R patients (Baruch et al, 2021; Davar et al, 2021) or healthy volunteers for improving the outcome of immunotherapy (Routy et al, 2023). According to the obtained results, the microbiota structure of R patients differed from the microbiota of NRs even before the FMT procedure was performed. Therefore, it is reasonable to assume that the R patient's microbiota responded to the donor feces in a way that enhanced antitumor immunity, whereas the NR patient's microbiota did not respond properly. Perhaps because of the absence (insufficient amount) of specific bacteria in the microbiota of the NRs, the donor feces were unable to produce such an improvement. Previous experiments in laboratory animals have suggested the possibility of supplementing cobalamin deficiency with fecal matter (Barnes & Fiala, 1959; Morgan et al, 1964). Absorption of donor fecal cobalamin is thought to enhance cytotoxic immunity, which may have a beneficial effect on immunotherapy outcomes. In addition, fecal corrinoids, which are inaccessible to humans, can be used by corrinoid auxotrophs in the gut for improved growth and metabolism. As the specter of colonizers was similar between R and NR patients in our previous analysis (Olekhnovich et al, 2023), it is possible that the FMT mechanism in this particular case is related to fecal cobalamin (or other metabolites) rather than donor-derived microbial colonization. On the other hand, the potential impact of donor microorganisms cannot be ignored. The state of the gut microbiota and the efficacy of immunotherapy may both be improved by the ability of donor microorganisms to restore lost ecological links, which are less degraded in Rs' compared with NRs'.

Based on the results obtained and data from the scientific literature, it is evident that fiber consumption may have a positive impact on melanoma immunotherapy outcomes (Spencer et al, 2021). However, further research in large patient cohorts is essential to identify the most effective fiber types and develop precise dosing regimens for clinical use. Although studies in laboratory animals using melanoma models have shown that the use of bifidobacteria and lactobacteria can enhance antitumor immunity (Sivan et al, 2015; Lee et al, 2021; Yoon et al, 2021; Si et al, 2022; Gao et al, 2023), it is important to note that Bifidobacterium spp. are also emerging as reproducible biomarkers of positive immunotherapy outcomes in studies and meta-analyses using stool metagenomes from melanoma patients (Lee et al, 2022; Olekhnovich et al, 2023; Zhao et al, 2023). Therefore, the incorporation of "classical probiotics" into immunotherapy regimens is promising on the one hand. On the other hand, the influence of commercial probiotics on immunotherapy outcomes has been reported to have negative effects (Spencer et al, 2021). It has been suggested that bifidobacteria may serve as markers of the "right" state of the human microbiota. The use of "classical" probiotics can be supplemented with butyrate-producing *Clostridium byturicum*, which increases the survival rate of metastatic renal cell carcinoma patients receiving immunotherapy (Dizman et al, 2022). Another potential probiotic candidate

for improving immunotherapy outcomes could be *Propionibacterium freudenreichii* because of its ability to produce cobalamin. Cobalamin-producing bacteria strengthen interactions within the gut microbiota and intestinal barrier tight junctions (Qi et al, 2023), which is beneficial for host resistance to pathogen infection and potentially beneficial for immunotherapy outcomes. However, the most promising prospects lie in the potential development of *F. prausnitzii*–based probiotics for clinical use (Khan et al, 2023). As noted above, this bacterium has been associated with the enhancement of antitumor immunity in numerous studies, including our own analysis. Thus, the use of other novel types of probiotics can be considered in the future as a potential direction for improving antitumor immunity.

Summarizing the results and hypothesis, it is evident that the mutualistic relationships of human gut microbes play a crucial role in the establishment aimed at enhancing antitumor immunity. Here are the key points of the conclusions drawn from the interpretation of the analysis results: (1) polysaccharide utilization and substrate sharing: Bacteroidetes species degrade the complex carbohydrates and produce substrates for other community members, including butyrate producers; (2) cobalamin sharing: cobalamin producers share cobalamin with Bacteroidetes and other cobalamin auxotrophs. Bacteroidetes, in turn, can help transport cobalamin within intestinal epithelial cells via extracellular vesicles, facilitating its distribution to other microbial communities and the host. This cooperative exchange ensures that essential nutrients are available to different members of the microbiota, thereby building community resilience; (3) cross-feeding in acetate/butyrate metabolism: butyrate producers use acetate derived from *Bifidobacterium* and/or obtained through the WL pathway for growth and butyrate production. This cross-feeding relationship results in increased production of both acetate and butyrate for the benefit of the host. The use of additional carbon sources such as $CO_2$ in the WL pathway may allow for the production of more acetate and butyrate, freeing up resources for the production of other important metabolites. Interestingly, this analysis suggests that it is not the ability to produce butyrate per se but rather the cross-feeding between bacteria associated with butyrate production that contributes to improved immunotherapy outcomes. This underscores the importance of microbial interactions in boosting immunity; (4) altruistic behavior: the microbiota beneficial for immunotherapy exhibit "altruistic" behavior, producing important metabolites such as amino acids and cobalamin and can distribute them to community members, which can improve the host's antitumor immune status. In contrast, the harmful microbiota behave "selfishly" and compete with the host for these essential resources, leading to weakened immunity.

In summary, our findings have advanced the understanding of the biological mechanisms of gut microbiome influence on the melanoma immunotherapy outcome and provided a foundation for further investigations aimed at enhancing immunotherapy efficacy through microbiome modulation. In summary, our findings have advanced the understanding of the biological mechanisms of gut microbiome influence on the melanoma immunotherapy outcome and provided a foundation for further investigations aimed at enhancing immunotherapy efficacy through microbiome modulation.

# Materials and Methods

### Metagenomic datasets, analysis, and data preprocessing

Sequencing data from gut metagenomic samples from melanoma patients were collected from seven published studies (Frankel et al, 2017; Gopalakrishnan et al, 2018; Matson et al, 2018; Baruch et al, 2021; Davar et al, 2021; Spencer et al, 2021; Lee et al, 2022). These data were preprocessed as follows. metagenomic data were quality-assessed using FastQC (https://github.com/s-andrews/FastQC). Technical sequences and low-quality bases were removed from the data using the Trimmomatic tool (Bolger et al, 2014). Human sequences present in the metagenomic samples were eliminated using bbmap (Bushnell, 2014) and the human genome GRCh37 (https://www.ncbi.nlm.nih.gov/genome/guide/human). All preprocessing computational steps were executed using the Assnake metagenomic pipeline (https://github.com/ASSNAKE). Detailed information on the characteristics of the metagenomic datasets and preprocessing statistics can be found in our previous study (Olekhnovich et al, 2023).

### Construction of a nonredundant catalog of MAGs

The sequencing data obtained from the preprocessing step were used to construct metagenomic contigs using the MEGAHIT assembler (Li et al, 2015). Contigs longer than 1,000 base pairs were retained for further analysis. The assembly results were then subjected to binning using two methods: MaxBin2 (Wu et al, 2016) and MetaBAT2 (Kang et al, 2019). DASTool was employed to create optimized, nonredundant bin sets for each sample (Sieber et al, 2018). To construct a nonredundant catalog of MAGs, the dRep tool was used with specific parameters: –completeness 75 and –contamination 10 –P_ani 0.9 –S_ani 0.98 (Olm et al, 2017). To assess the final quality of the resulting bin set, the CheckM framework was applied (Parks et al, 2015). Taxonomic annotation of the resulting MAGs catalog was performed using the Genome Taxonomy Database–Tk tool (Chaumeil et al, 2022; Parks et al, 2022). A phylogenetic tree incorporating all MAGs sequences was constructed utilizing the obtained CheckM AA marker set and the FastTree tool (Price et al, 2010). Multiple alignment of CheckM AA markers was performed using MUSCLE (Edgar, 2004). The EMPress tool (Cantrell et al, 2021), included in the QIIME2 framework (Bolyen et al, 2019), was used to visualize the phylogenetic tree of MAG biomarkers.

inStrain was used to obtain MAG abundance profiles (Olm et al, 2021). Using the abundance matrix of MAGs obtained in the previous step, the Shannon index was calculated using the diversity function in the vegan v2.6-4 package for GNU/R (https://github.com/vegandevs/vegan). Robust principal component analysis, implemented in the DEICODE package, was used for beta diversity assessment and two-dimensional visualization (Martino et al, 2019). The associations between experimental variables and microbial composition were evaluated using PERMANOVA with 10,000 permutations, implemented in the "adonis" function of the vegan v2.6-4 package, and the robust Aitchison distance calculated by DEICODE.

The results were visualized using the ggplot2 v3.4.2 library for GNU/R. (https://ggplot2.tidyverse.org).

### Strategy for discovering MAG biomarkers

The identification of MAG biomarkers associated with the immunotherapy outcome was performed in a similar manner to the previous article (Olekhnovich et al, 2023). In the first step, MAGs whose relative abundance was associated with immunotherapy outcome were identified using the Songbird (Morton et al, 2019). Absolute value threshold of the Songbird differential was >0.3. The log ratios of the selected MAGs were calculated using Qurro (Fedarko et al, 2020), whereas the statistical significance of the log ratios was accessed using the Wilcoxon rank-sum test implemented in the basic GNU/R function. The second step was to create a list of consistent MAG biomarkers using the following methodology: (1) microbial species associated with a positive immunotherapy outcome in more than one dataset were added to the list; (2) MAGs associated with a negative outcome in at least one dataset were excluded from the list of MAG biomarkers, regardless of the number of datasets in which they were associated with a positive outcome. The specific MAG biomarkers associated with adverse outcomes were also identified. OrthoFinder was used to construct a phylogenetic tree using MAG biomarker sequences. The resulting MAG biomarker phylogenetic tree was visualized using the EMPress tool. The ggplot2 v3.4.2 library for GNU/R was used to visualize the results.

Python libraries such as pandas (https://pandas.pydata.org), numpy (https://numpy.org), matplotlib (https://matplotlib.org), scikit-learn (https://scikit-learn.org), scipy (https://scipy.org) were used to build machine learning models based on MAG relative abundance values and log ratios with subsequent statistical evaluation by the Wilcoxon rank-sum test.

### Phylogenetic tree construction of *Faecalibacterium* and *Bifidobacterium* species

MAGs assigned to the genera *Faecalibacterium* (30 MAGs) and *Bifidobacterium* (12 MAGs) were used for phylogenetic analysis. Open reading frames and translated AA sequences from selected MAGs were predicted using Prodigal version 2.6.3 (Hyatt et al, 2010). Phylogenetic trees based on predicted sequences were reconstructed using OrthoFinder version 2.5.4 (Emms & Kelly, 2019) with default parameters. Genomes of species inhabiting the human gut were selected as references. The genome sequence of *G. vaginalis* strain UMB0386 (GenBank: https://www.ncbi.nlm.nih.gov/genbank/PKJK01000001.1) was used as an outgroup for *Bifidobacterium*, whereas the genome sequence of *S. variabile* strain DSM 15176 (GenBank: https://www.ncbi.nlm.nih.gov/genbank/ACBY02000001.1) was used as an outgroup for *Faecalibacterium*. Phylogenetic trees were visualized using the ggplot2 v3.4.2 and ggtree v3.6.2 packages for GNU/R (Xu et al, 2022). To further control the quality of the *Bifidobacterium* MAGs, the sequences were checked by GUNC v1.0.5 (Orakov et al, 2021) to filter out chimeric genomes based on the "pass.GUNC" column in the gunc_output file.

### Functional profiling of MAG biomarkers

To investigate the presence of CAZy in MAGs, we performed a series of bioinformatic analyses. AA sequences, predicted by Prodigal

version 2.6.3 (Hyatt et al, 2010), were aligned against bacterial protein sequences from the CAZy database (http://www.cazy.org) (Drula et al, 2022) and the KEGG database (https://www.genome.jp/kegg) (Kanehisa et al, 2017) using the blastp mode of DIAMOND (Buchfink et al, 2021) (version 2.0.15) with a stringent threshold of 80% identity and 80% query coverage. In addition, we used the gapseq method (Zimmermann et al, 2021) in conjunction with the MetaCyc database (https://metacyc.org) (Caspi et al, 2020) for further functional annotation of MAG biomarkers.

Nonmetric multidimensional scaling using the Bray–Curtis dissimilarity metric was used to visualize the functional data in two dimensions (https://github.com/vegandevs/vegan). To measure dissimilarities in the functional profiles among MAG biomarkers, we performed PERMANOVA using the "adonis" function from the vegan package and the Bray–Curtis metric. Differences in the number of functional categories between biomarker sets were assessed using the Wilcoxon rank-sum test with FDR correction for multiple testing. Furthermore, differences in functional content between the MAGs groups were determined using one-sided Fisher exact tests with FDR correction, implemented in GNU/R.

To detect differences in the KEGG/MetaCyc gene sets between the MAGs groups, we used GSEA from Bioconductor's "piano" package (Väremo et al, 2013). Specifically, we used the "reporter feature algorithm" with a gene set significance threshold of adj. $P <$ 0.01 and gseaParam = 1. FDR-corrected $P$-values derived from the Fisher exact test were used as input data for the GSEA analysis. Only genes with uncorrected $P$-values < 0.5 were included in the analysis. Results were visualized using the ggplot2 v3.4.2 and pheatmap v1.0.12 (https://github.com/raivokolde/pheatmap) libraries for GNU/R.

Metabolic pathways responsible for acetate, butyrate, and propionate production were focused for the additional analysis. For acetate, we considered six possible biosynthesis pathways, including the WL pathway and a recently discovered pathway involving succinyl-CoA:acetate CoA-transferase and succinyl-CoA synthetase (Koh et al, 2016; Esposito et al, 2019; Zhang et al, 2021). Meanwhile, for butyrate and propionate, we explored four and three possible synthetic pathways, respectively (Vital et al, 2014; Frolova et al, 2022) (refer to Table S15 for details). We assembled a reference catalog of gene products for each pathway, resulting in 4,563 AA sequences for acetate pathways, 2,744 for butyrate pathways, and 415 for propionate pathways. Subsequently, DIAMOND (version 2.0.15) blastp searches (Buchfink et al, 2021) and the program gapseq (v1.1) (Zimmermann et al, 2021) with default parameters were used to validate the presence of these pathways. In addition, we used gapseq profiles and flux balance analysis to predict AAs and SCFA consumers/producers (Gelius-Dietrich et al, 2013; Zimmermann et al, 2021).

## Data Availability

In this study, we used open access data from the NCBI-EBI Sequence Read Archives, identified by the following BioProjects accession numbers: PRJNA397906, PRJEB22893, PRJNA399742, PRJNA678737, PRJNA672867, PRJNA770295, and PRJEB43119. Extensive results from our project are detailed in the article text, along with supporting materials. We have also provided a catalog of metagenome-assembled genomes (MAGs), taxonomic annotation results, phylogenetic trees, and Qurro and EMPress profiles for the QIIME2 viewer. These resources have been made available through the figshare service, which can be accessed via the following link: 10.6084/m9.figshare.24146913.v5.

## Supplementary Information

## Acknowledgements

Financial support for this study was provided by the Russian Science Foundation under the grant number 22-75-10029, available at https://rscf.ru/project/22-75-10029/. This work was performed using the core facilities of the Lopukhin FRCC PCM "Genomics, proteomics, metabolomics" (http://rcpcm.org/nauchnye-issledovanija/centr-kollektivnogo-polzovaniia/). We also thank Roman Yunes for his assistance in editing the English text.

### Author Contributions

NV Zakharevich: formal analysis, validation, visualization, and writing—original draft.
MD Morozov: data curation and writing—original draft.
VA Kanaeva: visualization.
MS Filippov: software, validation, visualization, and methodology.
TI Zyubko: conceptualization, validation, visualization, and methodology.
AB Ivanov: validation, visualization, and writing—review and editing.
VI Ulyantsev: conceptualization, project administration, and writing—review and editing.
KM Klimina: formal analysis, funding acquisition, and writing—review and editing.
EI Olekhnovich: conceptualization, resources, data curation, formal analysis, supervision, funding acquisition, validation, investigation, visualization, methodology, writing—original draft, and project administration.

### Conflict of Interest Statement

The authors declare that they have no conflict of interest.

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
