## [Reviewer comments · Life Science Alliance]

Life Science Alliance

Systemic Metabolic Depletion of Gut Microbiome Undermines Responsiveness to Melanoma Immunotherapy

Natalia V. Zakharevich, Maxim D. Morozov, Vera A. Kanaeva, Mikhail S. Fillipov, Tatyana I. Zyubko, Artem B. Ivanov, Vladimir I. Ulyantsev, Ksenia M. Klimina and Evgenii I. I. Olekhovich

DOI: <https://doi.org/10.26508/lsa.202302480>

Corresponding author(s): Dr. Evgenii I. I. Olekhovich (Lopukhin Federal Research and Clinical Center of Physical-Chemical Medicine of Federal Medical Biological Agency)

Review Timeline:

Submission Date:	2023-11-14
Editorial Decision:	2023-12-18
Revision Received:	2024-01-24
Editorial Decision:	2024-02-07
Revision Received:	2024-02-09
Accepted:	2024-02-12

Transaction Report:

December 18, 2023

Re: Life Science Alliance manuscript #LSA-2023-02480-T

Dr. Evgenii I. I. Olekhovich
Lopukhin Federal Research and Clinical Center of Physical-Chemical Medicine of Federal Medical Biological Agency
Pirogovskaya 1a
Moscow 123456
Russian Federation

Dear Dr. Olekhovich,

Thank you for submitting your manuscript entitled "Computational model of intestine microbiome influence on melanoma immunotherapy outcome" to Life Science Alliance. The manuscript was assessed by expert reviewers, whose comments are appended to this letter. We invite you to submit a revised manuscript addressing the Reviewer comments.

Thank you for this interesting contribution to Life Science Alliance. We are looking forward to receiving your revised manuscript.

Sincerely,

B. MANUSCRIPT ORGANIZATION AND FORMATTING:

Reviewer #1 (Comments to the Authors (Required)):

Zakharevich and coauthors aggregated fecal microbiome profiles from 7 different published studies encompassing 680 individual samples (from 374 responders and 306 nonresponders) and re-analyzed them on a unified computational pipeline. This work extends a recent study of theirs (in mSystems) in which some of the same sample-sets were analyzed.

They now extend their prior work by (a) adding datasets and more importantly (b) assembling, in silico, metagenome-assembled genomes (MAGs) which represent 'complete' genomes of ~1400 organisms present within the samples. They then identified MAG-species biomarkers of response vs non-response to immunotherapy. They also drill down to biochemical pathways (KEGG- and MetaCyc-annotated features) in identifying potential candidates. This analysis revealed several amino-acid production/utilization pathways and cobalamin metabolism as potential biomarkers of response to checkpoint blockade.

This analysis is very welcome in the field as there has been a lot of heterogeneity in the associations reported across the different published studies, and so finding some shared biomarkers across datasets is of value. The main limitation of the study is that it lacks validation approaches (such as cross-validation, or designating some datasets as discovery sets and leaving others for validation) and as such it is susceptible to over-fitting. Nevertheless, it is still a valuable contribution and will be of particular interest in the field of immune-checkpoint-blockade/immunotherapy

Specific comments:

Figure 3A top panels: not clear to me what is plotted here. It is stated that what is plotted are log ratios of MAGs between responders and non-responders. If that is the case, why are there two different values, one for responders and one for non-responders?

Figure 3A lower panels: also not clear to me what is plotted here. The fecal transplant recipients could have been serially sampled, the donors were sampled only at the time of the donation and not multiple times during the recipients' treatment courses. So what does it mean for the donors' curves to extend across time?

The paragraph on p 5 that starts with "Biomarker identification..." has many phrases that conflate correlation/association with causal relationships between variables. E.g "characteristics that allow patients to respond", "beneficial donors have the capacity to improve the condition of recipients" "observed to exacerbate." Most of the studies were observational in nature, and the FMT studies did not include randomization to control arms, so no causal inferences/claims should be made.

Figure 2b: not clear what the reader should interpret from this. I assume that rows are samples and columns are MAGs? And the color coding within the heatmap represents abundance? Please clarify in the figure caption and provide a legend of the color scale.

Figure 3B-D & Figure 4B and Figure 5 and the accompanying text uses terminology for status "positive" and "negative" biomarkers. Is the reader to assume that 'positive' is synonymous with "associated with responder" and 'negative' means 'associated with non-responder'? Later (page 7) we encounter the term "R biomarkers." If so, consider using uniform terminology across the figures & manuscript, or defining these terms.

Figure 4A: please orient the reader as to how to interpret the "GSEA Statistic axis?" What does a higher or lower value imply?

Figure 6: fonts are illegibly small. This reviewer had to view PDF at 500% on a large monitor to read them. Please print the figures on standard-size paper and verify font legibility.

Minor points

Page 2: pull back "sensational" to a more conservative statement

Page 5: typo: "... in comparison to o FMT..."

Page 14: typo "a crucial role in esSummarizing"

Page 9: "It is well established that SCFAs....". It is not so well-established. Please pull this statement back to something like several studies have reported or something more conservative.

The text contains several colloquial phrases that should be formalized. E.g. "allows you to significantly improve" and many contractions e.g. "What's more", "Here's an overview of our overall analysis protocol"

Page 10: move the commentary about Anna Karenina principle ot Discussion section

Figure pages were not labeled with "Figure 1", "Figure 2" etc which makes it confusing to navigate the manuscript.

Line numbers would have been helpful to refer to specific spots within the manuscript.

Reviewer #2 (Comments to the Authors (Required)):

Drs. Zakharevich, Olekhnovich and colleagues have prepared a very nicely-written, compelling examination of 7 previously published studies where patients underwent treatment with immunotherapy for metastatic melanoma. While other groups have reported similar efforts re-examining many prior studies in a meta-analysis, the major novelty and strength of this study is in the methodology. In this study, the authors construct Metagenome-Assembled Genomes (MAGs) for each individual sample, and interestingly used this method to achieve distinct high-resolution (strain-level) taxonomical associations, such as *Faecalibacterium* spp. I think this study will be of interest to a broad audience. I do have some minor comments that I think addressing would result in a stronger manuscript.

- Can the authors quantify the information gained by using this method of assembling MAGs, as opposed to analyzing without assembly? Should MAG-based analysis be utilized as an adjunct analysis strategy in addition to MAG-independent analysis, or is it superior and can be utilized as an analysis strategy that replaces MAG-independent analysis?

- The authors identify strains of *Faecalibacterium* spp. with distinct associations to response, which is very interesting and could shed light on potential mechanisms of action. Can they describe how these strains with distinct response associations differ in their genetic features?

- The authors found that *Akkermansia* was associated with negative outcomes in 4 studies. *Akkermansia* has also been identified as being associated with outcomes in other studies. Recently, *Akkermansia* genetic heterogeneity has been examined and found to be composed of as many as 5 subspecies (PMID 34261503). Can the authors identify these subspecies in with their method, and comment on whether these subspecies have distinct associations with outcomes?

Dear Editor and Reviewers,

Much thanks for frank evaluation of our work, and for the time you have spent when reviewing our manuscript. We agree with nearly all of your remarks and have made appropriate modifications of the primary version (see "article_latex.pdf" and "change.pdf" file attached in the supplementary section).

Sincerely,
Eugene Olekhovich on behalf of the authors

Reviewer #1 (Comments to the Authors (Required)):

General comment: Zakharevich and coauthors aggregated fecal microbiome profiles from 7 different published studies encompassing 680 individual samples (from 374 responders and 306 nonresponders) and re-analyzed them on a unified computational pipeline. This work extends a recent study of theirs (in mSystems) in which some of the same sample-sets were analyzed.

They now extend their prior work by (a) adding datasets and more importantly (b) assembling, in silico, metagenome-assembled genomes (MAGs) which represent 'complete' genomes of ~1400 organisms present within the samples. They then identified MAG-species biomarkers of response vs non-response to immunotherapy. They also drill down to biochemical pathways (KEGG- and MetaCyc-annotated features) in identifying potential candidates. This analysis revealed several amino-acid production/utilization pathways and cobalamin metabolism as potential biomarkers of response to checkpoint blockade.

This analysis is very welcome in the field as there has been a lot of heterogeneity in the associations reported across the different published studies, and so finding some shared biomarkers across datasets is of value. The main limitation of the study is that it lacks validation approaches (such as cross-validation, or designating some datasets as discovery sets and leaving others for validation) and as such it is susceptible to over-fitting. Nevertheless, it is still a valuable contribution and will be of particular interest in the field of immune-checkpoint-blockade/immunotherapy.

Answer:

We appreciate the reviewer's appreciation of our work. Really, identifying clear and reproducible stool microbial markers is a significant challenge in modern medical science. I think that almost all studies in this area are susceptible to overfitting. Otherwise, prediction systems based on human microbiota data would be widespread right now.

In my opinion, there are two main reasons for this:

(1) Microbial diversity varies greatly between individuals due to objective factors such as lifestyle, dietary habits, geographic location, and genetic characteristics. The incompleteness of databases for taxonomic annotation exacerbates this problem, leading to extensive distortions in the determination of the microbial composition of samples.

(2) The sample sets are too small to control for the significant level of metagenomic data variance, which is further complicated by the issue of batch effects between different

data sets. In other words, the results obtained from one sample may not necessarily apply to another dataset.

Possible solutions to these issues:

(1) To address the initial issue, we avoided utilizing database-based methods like metaphlan (<https://github.com/biobakery/MetaPhlAn>). Instead, we extracted diversity directly from the data through MAGs assembling. However, the set of obtained genomes is specific only to this data. It is important to note that other datasets may be characterized by the presence of completely different genomes. It is improbable that a complete collection of microbial genomes from the intestines will be obtained for the entire human population in the near future. Therefore, it is improbable that it will be possible to predict the response to immunotherapy analytically, i.e., using a precise set of specific microbial species. It is improbable that a complete collection of microbial genomes from the intestines will be obtained for the entire human population in the near future. Therefore, it is improbable that it will be possible to predict the response to immunotherapy “analytically”, i.e., using a precise set of specific microbial species. However, understanding the mechanism of the immunomodulatory nature of the intestinal microbiota can allow for the utilization of functional pathways common in different biomarker organisms. In essence, the significance lies not in the microbes themselves, but rather in the functions they carry out. In other words, researchers can use a “heuristic” approach to solve this problem.

(2) Based on widely accepted principles of machine learning, it may appear that increasing the number of metagenomes from datasets generated by various sequencing runs would improve prediction power of models. However, a model trained on such data will not be effective on a dataset that was not used for training. In other words, the resulting model will only be applicable to the specific datasets used for training and cannot be used to characterize the overall plurality of human stool microbiomes, i.e., overfitting. To support this thesis, the cross-validation experiment was carried out as described below. For this purpose, generally accepted approaches for this field were used. Taxonomic annotation was obtained through the use of Metaphlan4, and we employed the Random Forest (RF) algorithm to construct predictive models. This experiment employs two different cross-validation approaches: out-of-fold 6/1 cross-validation (using 6 datasets for training and 1 for cross-validation) and random shuffle cross-validation (using 70% random samples for training and 30% for testing). The obtained results are presented in Figure A1.

Figure A1-R1. Cross-validation results for RF prediction models. The x-axis displays model quality metrics, while the y-axis shows the corresponding metric values. The different cross-validation methods described in the text are highlighted in different colors: 1 - red, 2 - blue.

According to the Figure A1-R1 data, models trained on 6 datasets were unable to accurately predict the response to melanoma immunotherapy on the remaining dataset. However, data shuffling solved this problem. The RF model performs well when it receives information about the entire dataset. In other words, models trained on a large amount of existing metagenomic data will not be effective for new sets, making it impractical to develop such predictive models for implementation in practice.

Of course, batch effects control methods can be used to solve this problem. However, their application may also lead to distortions. A model trained on a dataset with a corrected batch effect may still lack predictive power on a third-party dataset that did not participate in the correction. Certainly, without evidence, this thesis does not hold much power and may be incorrect. However, further evidence may be necessary to fully address this issue, and it may be beyond the scope of our current work.

The solution to the second problem involves using log ratios, which more closely match the structure of metagenomic data, rather than relative abundances or read counts [Morton et al., 2019; Fedarko et al., 2020; Rahman et al., 2023]. In the article, we cross-validated the obtained biomarkers using a strong out-of-dataset 6/1 method, as described above, with different machine learning approaches. The log ratios yielded superior results compared to relative abundances (refer to Figure S2 and pages 5-6 lines 171-199 in 'article_latex.pdf' file). It is important to note that the MAG-biomarkers obtained are relevant only in the context of studying biological phenomena in this study and not for the development of clinically relevant systems for predicting response to immunotherapy. As the first problem has not yet been solved, any development in this direction can only be considered as research, but not practical.

Specific comments:

1) Figure 3A top panels: not clear to me what is plotted here. It is stated that what is plotted are log ratios of MAGs between responders and non-responders. If that is the case, why are there two different values, one for responders and one for non-responders?

Answer:

The log ratios were calculated as follows: 1) Songbird differentials indicating the association power with immunotherapy responsiveness were calculated for each dataset separately. 2) Select MAGs with an absolute value of Songbird differential > 0.3 in each sample. 3) Calculate $\log(\text{sum of relative abundances of numerators} / \text{sum of relative abundances of denominators})$ for each sample (according to Qurro [Fedarko et al., 2020]), where numerators are MAGs with Songbird differential > 0.3 and denominators are MAGs with Songbird differential < -0.3. Each point on the boxplots of Figure 3A represents one sample. The value obtained indicates the difference in the presence of numerators and denominators in the sample. An increase in the resulting numerator shows a shift towards a higher abundance of microbes associated with a positive immunotherapy outcome. Conversely, a decrease in the log ratio indicates a higher abundance of microbes associated with a negative immunotherapy outcome. The biomarker selection protocol is described in the

Results section on page 4 lines 122-136 in 'article_latex.pdf' file, and in the 'Materials and Methods' section on page 15 lines 594-609 in 'article_latex.pdf' file. It is agreed that the descriptions of the log-ratio plots (Figure 3A and Figure 3B) were unclear, but we have supplemented them to improve understanding. The captions for Figure 3A have also been edited to enhance clarity (page 17 lines 688-705 in 'article_latex.pdf' file, page 23 lines 954-975 in 'changes.pdf' file).

2) Figure 3A lower panels: also not clear to me what is plotted here. The fecal transplant recipients could have been serially sampled, the donors were sampled only at the time of the donation and not multiple times during the recipients' treatment courses. So what does it mean for the donors' curves to extend across time?

Answer:

This is a continuation of the previous comment. Log ratios between R and NR MAGs were calculated for each metagenome. Then, the samples were combined based on their affiliation with specific recipients. The figure demonstrates changes in the recipients' microbiomes over time expressed by log ratios, with the 0 time point on the x-axis representing the state of the microbiota before the FMT procedure. The FMT data has been separated from the non-FMT data in the figure. The captions for the drawing have been expanded to improve clarity. Please refer to page 17 lines 688-705 in 'article_latex.pdf' or page 23 lines 954-975 in 'changes.pdf' file for more information.

3) The paragraph on p 5 that starts with "Biomarker identification..." has many phrases that conflate correlation/association with causal relationships between variables. E.g "characteristics that allow patients to respond", "beneficial donors have the capacity to improve the condition of recipients" "observed to exacerbate." Most of the studies were observational in nature, and the FMT studies did not include randomization to control arms, so no causal inferences/claims should be made.

Answer:

Our reasoning was overly optimistic. This section of the article contains more speculation than identified facts. It has been revised according to the reviewer's comment (page 4 lines 143-153 in 'article_latex.pdf' file or page 6 lines 221-243 in 'changes.pdf').

4) Figure 2b: not clear what the reader should interpret from this. I assume that rows are samples and columns are MAGs? And the color coding within the heatmap represents abundance? Please clarify in the figure caption and provide a legend of the color scale.

Answer:

The figure and captions were edited to address the reviewer's comments. (Figure 2B; page 17 lines 669-687 in 'article_latex.pdf' or pages 22-23 lines 931-953 in 'changes.pdf').

5) Figure 3B-D & Figure 4B and Figure 5 and the accompanying text uses terminology for status "positive" and "negative" biomarkers. Is the reader to assume that 'positive' is synonymous with "associated with responder" and 'negative' means 'associated with non-responder.? Later (page 7) we encounter the term "R biomarkers." If so,

consider using uniform terminology across the figures & manuscript, or defining these terms.

Answer:

We agree with inconsistent terminology in the text. This has been fixed.

**6) Figure 4A: please orient the reader as to how to interpret the "GSEA Statistic axis?"
What does a higher or lower value imply?**

Answer:

The figure captions have been revised to address the reviewer's comment.

7) Figure 6: fonts are illegibly small. This reviewer had to view PDF at 500% on a large monitor to read them. Please print the figures on standard-size paper and verify font legibility.

Answer:

It has been acknowledged that Figure 6 was removed from the article. Instead, Figure S2 was added and enlarged.

Minor points:

8) Page 2: pull back "sensational" to a more conservative statement

9) Page 5: typo: "... in comparison to o FMT..."

10) Page 14: typo "a crucial role in esSummarizing"

11) Page 9: "It is well established that SCFAs....". It is not so well-established. Please pull this statement back to something like several studies have reported or something more conservative.

12) The text contains several colloquial phrases that should be formalized. E.g. "allows you to significantly improve" and many contractions e.g. "What's more", "Here's an overview of our overall analysis protocol"

13) Page 10: move the commentary about Anna Karenina principle ot Discussion section

14) Figure pages were not labeled with "Figure 1", "Figure 2" etc which makes it confusing to navigate the manuscript.

15) Line numbers would have been helpful to refer to specific spots within the manuscript.

All minor points have been corrected.

References:

- 1) Morton, James T., et al. "Establishing microbial composition measurement standards with reference frames." *Nature communications* 10.1 (2019): 2719
- 2) Fedarko, Marcus W., et al. "Visualizing omic feature rankings and log ratios using Qurro." *NAR genomics and bioinformatics* 2.2 (2020): lqaa023.
- 3) Rahman, Gibraan, et al. "BIRDMAN: A Bayesian differential abundance framework that enables robust inference of host-microbe associations." *bioRxiv* (2023): 2023-01.

Reviewer #2 (Comments to the Authors (Required)):

General comment: Drs. Zakharevich, Olekhovich and colleagues have prepared a very nicely-written, compelling examination of 7 previously published studies where patients underwent treatment with immunotherapy for metastatic melanoma. While other groups have reported similar efforts re-examining many prior studies in a meta-analysis, the major novelty and strength of this study is in the methodology. In this study, the authors construct Metagenome-Assembled Genomes (MAGs) for each individual sample, and interestingly used this method to achieve distinct high-resolution (strain-level) taxonomical associations, such as *Faecalibacterium* spp. I think this study will be of interest to a broad audience. I do have some minor comments that I think addressing would result in a stronger manuscript.

Answer:

I thank the reviewer for acknowledging our work.

Specific comments:

1) Can the authors quantify the information gained by using this method of assembling MAGs, as opposed to analyzing without assembly? Should MAG-based analysis be utilized as an adjunct analysis strategy in addition to MAG-independent analysis, or is it superior and can be utilized as an analysis strategy that replaces MAG-independent analysis?

Answer:

Quantifying MAG-based and common methods appears to be a challenging task. Reads were mapped to a full-length genome catalog in our study, while one of the community-accepted common MetaPhlan4 (<https://huttenhower.sph.harvard.edu/metaphlan>), for instance, used unique markers for species quantification. Usually, common methods typically rely on databases, and the quality of these databases affects the classification accuracy. Thus, comparing the amount of mapped reads is not meaningful, but we can attempt to compare the results of these methods. For a clearer demonstration, we used the log-ratio approach described in our article. We collected four additional datasets and performed all the analysis steps described in this article which is a continuation of our ongoing work. Figure A1-R2-A provides a visual representation of the results. We ensured that the resulting set of marker MAGs worked perfectly in out-of-dataset cross-validation: all minus 1 datasets were used for training and 1 remaining dataset was used for testing. Using MetaPhlan4 annotation results with the latest version of the database, a statistically significant decrease in classification quality was observed when performing the same manipulations on the data (refer to Figure A1-R2-B; Wilcoxon rank-sum test, $p < 0.01$).

I think that MetaPhlan4 can be used to obtain primary results quickly, but it does not account for intraspecific variability. However, StrainPhlan

(<http://segatalab.cibio.unitn.it/tools/strainphlan>) can be used to study this variability without assessing functional potential. The HUMANN2 (<https://huttenhower.sph.harvard.edu/humann2>) pipeline, which is produced by the Curtis Huttenhower and Nicola Segata labs and belongs to the same family of algorithms, associates functions with taxonomy in a limited way also based on the database. In contrast, The MAG-based strain profiling approach, although more challenging to implement and interpret, allows for a deeper understanding of biological processes by delving into metagenomic data and taking into account the intraspecific variability of microorganisms. Moreover, specific statistical procedures are required for proper execution of this approach. However, the results obtained may be less transferable to other datasets. Creating an exhaustive representative catalog of genomes would be the ideal option to solve such complex research problems. It is hoped that future studies will achieve this.

Figure A1-R2. Cross validation results of log-regression prediction models. To train the models were used (A) log ratios calculated based on MAG-markers profiles or (B) log ratios calculated based on MetaPhiAn2 profiles.

2) The authors identify strains of *Faecalibacterium* spp. with distinct associations to response, which is very interesting and could shed light on potential mechanisms of action. Can they describe how these strains with distinct response associations differ in their genetic features?

Answer:

We performed such a test and did not identify any gene groups distinguishing marker MAGs from neutral MAGs. We think that important functions can be within mobile elements or other difficult sites that can be lost in the binning procedure. Further investigation into the variability of *Faecalibacterium* spp. in the context of immunotherapy responsiveness could be a valuable research avenue. However, to answer this question, a special analysis must be designed that would be worthy of its own publication. If time and resources will allow, we could do such a study. Also, we are open to co-operation. If a reviewer is interested in this research, we are happy to collaborate.

3) The authors found that Akkermansia was associated with negative outcomes in 4 studies. Akkermansia has also been identified as being associated with outcomes in other studies. Recently, Akkermansia genetic heterogeneity has been examined and found to be composed of as many as 5 subspecies (PMID 34261503). Can the authors identify these subspecies in with their method, and comment on whether these subspecies have distinct associations with outcomes?

Answer:

In the metagenomes analyzed in our work, we found 6 MAGs belonging to the genus *Akkermansia*: ERR6243878.3, SRR5930509.003_sub, ERR6279660.003, SRR13022350.13, ERR6279626.93 and ERR2162205.37. Only one of them was included in the biomarkers – ERR6243878.3 (NR biomarker). In order to classify all 6 MAGs according to the five candidate species proposed in the article PMID 34261503 (SGB9223, SGB9224, SGB9227, SGB9228 and SGB9226 – belonging to the type strain *A. muciniphila* ATCC BAA-835), we carried out a phylogenetic analysis. To reconstruct the phylogenetic tree, we took *Akkermansia* isolates from NCBI classified by the authors of the article (accession number is given by the authors of the article in the Supplementary files). Since there were only 2, 4 and 9 such sequences (genomes assigned to a given candidate species) for the candidate species SGB9227, SGB9224 and SGB9228 respectively, for the candidate species SGB9223 and SGB9226 (for which more sequences were available), we took 10 “reference” sequences each from NCBI.

Figure A2-R2. Phylogenetic tree reconstructed using OrthoFinder orthogroups. Our MAGs are highlighted in color: green - not included in the biomarkers, blue - included. Candidate species numbers are in bold.

As can be seen from the resulting phylogenetic tree, 6 *Akkermansia* MAG's that we identified were distributed among the proposed candidate species. MAG ERR6243878.3 (NR biomarker) was included in the same group with the candidate species SGB9223. The rest MAGs were distributed as follows: ERR6279626.93 and ERR2162205.37 – SGB9224, SRR5930509.003_sub – SGB9227, ERR6279660.003 – SGB9228 and SRR13022350.13 – SGB9226.

Since *Akkermansia* strains differ in their potential to produce vitamin B₁₂ (cobalamin), the authors of the article PMID 34261503 characterized capabilities of the candidate species to synthesis this important cofactor. We followed the authors of the article and, like them, we checked our *Akkermansia* MAGs for the presence of corrin ring biosynthesis genes (*cbiKP*, *cbiL(cobI)*, *cbiC*, *cbiD*, *cbiE/cbiT*, *cbiF(cobM)*, *cbiA*, *cobO*) – as a proxy for vitamin B12 synthesis capability. In four MAGs ERR6243878.3 (SGB9223), SRR13022350.13

(SGB9226), ERR6279626.93 and ERR2162205.37 (SGB9224) we did not find a single gene for the biosynthesis of the corrin ring. And in the next two MAGs: SRR5930509.003_sub and ERR6279660.003 – all eight genes were found, and these MAGs belong to candidate species SGB9227 and SGB9228, respectively. Our results completely coincide with the patterns discovered by the authors of the article PMID 34261503.

The fact that genes involved in the synthesis of vitamin B₁₂ were found in MAGs that were not included in the biomarkers (SRR5930509.003_sub and ERR6279660.003) and were not found in the NR biomarker MAG ERR6243878.3 does not contradict our logic. According to our study, the presence of genes associated with vitamin B₁₂ synthesis is associated with response to immunotherapy and the statistical analysis revealed an increase in the number of cobalamin biosynthesis genes in positive (R) biomarkers, but not in negative (NR) ones. Which confirms one of our conclusions – altruistic behavior: microbiota beneficial for immunotherapy exhibit "altruistic" behavior, producing important metabolites such as amino acids and cobalamin, and can distribute them to community members, which can improve the host's anti-tumor immune status. In contrast, the harmful microbiota behave "selfishly" and compete with the host for these essential resources, leading to weakened immunity. Also the loss of vitamin B₁₂ synthesis genes in a NR biomarker MAG ERR6243878.3 may be metabolically beneficial for it – if we take into account the hypothesis Karcher and co-authors (PMID 34261503), according to which in this case it is necessary to take into account the possibility of syntrophic interaction of this strain with other species – in our case, components of the gut microbiota of the non-responder.

February 7, 2024

RE: Life Science Alliance Manuscript #LSA-2023-02480-TR

Dr. Evgenii I. I. Olekhovich
Lopukhin Federal Research and Clinical Center of Physical-Chemical Medicine of Federal Medical Biological Agency
Pirogovskaya 1a
Moscow 123456
Russian Federation

Dear Dr. Olekhovich,

Thank you for submitting your revised manuscript entitled "Systemic Metabolic Depletion of Gut Microbiome Undermines Melanoma Immunotherapy Responsiveness". We would be happy to publish your paper in Life Science Alliance pending final revisions necessary to meet our formatting guidelines.

- please address the Reviewers' remaining comments. Numbering the figures at this stage is not needed, but Reviewer 1's recommendation is a good one for future submissions.
- please be sure that the authorship listing and order is correct
- please add the Twitter handle of your host institute/organization as well as your own or/and one of the authors in our system
- please move your main, supplementary figure, and table legends to the main manuscript text after the references section
- we encourage you to revise the figure legend for Figure 3 such that the figure panels are introduced in alphabetical order
- please rename the Conclusions section to Discussion

A. FINAL FILES:

B. MANUSCRIPT ORGANIZATION AND FORMATTING:

Sincerely,

Reviewer #1 (Comments to the Authors (Required)):

This revision has been very responsive to my critiques and it is considerably improved.

Minor points

Fig 2B - typo in legend. Absence vs "abcense" and also, the caption to 2A has two inaccuracies: One is it says that legend color bars are on the right but they are on the left. Second is that it says that "MAG abundances" are plotted on the y-axis. I think it is MAGs themselves that are on the y axis, and presence/absence (or abundance) is in the cells of the heatmap, but it would not be accurate to say that the abundance represented along the length of the y-axis.

A general logistical comment to both to the authors and editors: The figures themselves are not numbered, which makes it very hard to navigate the manuscript. The only way to identify which figure is which is to serially count from the beginning, each time the text refers the reviewer to one fo the figures. And if the figures are printed, they can easily get out of order. If this were somewhat manageable for the main figures, it becomes too much for the 10-figures supplement. In future manuscripts, I encourage the authors to label figures ("Figure 1" etc) and I encourage the editorial staff insist on this before sending out manuscripts for review. I commented on this briefly in first round and this revision came back again without numbered figures so I appreciate the authors' patience with my lengthy explanation for why this is so helpful to the reviewer.

Reviewer #2 (Comments to the Authors (Required)):

The authors have been very responsive to reviewer comments, performing additional analyses, but most of these are not included in the revised manuscript. I personally feel that including these would enhance their manuscript, but I leave it to them (and the editors) to decide. I have no concerns with the revised manuscript.

Dear Editor and Reviewers,

Thank you for your careful review and additional comments. In the future, we will pay more attention to the numbering of lines and figures.

Sincerely,
Eugene Olekhovich on behalf of the authors

Reviewer #1 (Comments to the Authors (Required)):

This revision has been very responsive to my critiques and it is considerably improved.

We appreciate the feedback provided by the first reviewer. His input has been valuable in improving the manuscript.

Minor points

Fig 2B - typo in legend. Absence vs "abcense" and also, the caption to 2A has two inaccuracies: One is it says that legend color bars are on the right but they are on the left. Second is that it says that "MAG abundances" are plotted on the y-axis. I think it is MAGs themselves that are on the y axis, and presence/absence (or abundance) is in the cells of the heatmap, but it would not be accurate to say that the abundance represented along the length of the y-axis.

The text of the article has been appropriately modified.

A general logistical comment to both to the authors and editors: The figures themselves are not numbered, which makes it very hard to navigate the manuscript. The only way to identify which figure is which is to serially count from the beginning, each time the text refers the reviewer to one of the figures. And if the figures are printed, they can easily get out of order. If this were somewhat manageable for the main figures, it becomes too much for the 10-figures supplement. In future manuscripts, I encourage the authors to label figures ("Figure 1" etc) and I encourage the editorial staff to insist on this before sending out manuscripts for review. I commented on this briefly in first round and this revision came back again without numbered figures so I appreciate the authors' patience with my lengthy explanation for why this is so helpful to the reviewer.

We apologize to the reviewer for the lack of numbered illustrations. We've become accustomed to what the article's loading system is responsible for numbering both the text and figures. Additional materials and main figures have been numbered by us in the file names, which apparently wasn't enough. We regret the inconvenience to the reviewer. We'll be more careful with future article submissions.

Reviewer #2 (Comments to the Authors (Required)):

The authors have been very responsive to reviewer comments, performing additional analyses, but most of these are not included in the revised manuscript. I personally feel that including these would enhance their manuscript, but I leave it to them (and the editors) to decide. I have no concerns with the revised manuscript.

The questions and issues raised by the reviewer are very interesting and, in our opinion, very important, and they deserve more in-depth consideration and research. We are preparing a continuation of our work in which we will explore the issues raised in greater depth and detail.

February 12, 2024

RE: Life Science Alliance Manuscript #LSA-2023-02480-TRR

Dr. Evgenii I. I. Olekhovich
Lopukhin Federal Research and Clinical Center of Physical-Chemical Medicine of Federal Medical Biological Agency
Pirogovskaya 1a
Moscow 123456
Russian Federation

Dear Dr. Olekhovich,

Thank you for submitting your Research Article entitled "Systemic Metabolic Depletion of Gut Microbiome Undermines Responsiveness to Melanoma Immunotherapy". It is a pleasure to let you know that your manuscript is now accepted for publication in Life Science Alliance. Congratulations on this interesting work.

DISTRIBUTION OF MATERIALS:

Again, congratulations on a very nice paper. I hope you found the review process to be constructive and are pleased with how the manuscript was handled editorially. We look forward to future exciting submissions from your lab.

Sincerely,
